# CONTINUAL PRE-TRAINING FOR HALLUCINATION REDUCTION

## ABSTRACT

Hallucinations, where model outputs contradict or cannot be verified by the provided evidence, remain a central obstacle to the reliable use of large language models (LLMs). Thus, it remains to be seen how hallucination can be decreased via sophisticated training methods. The theory that the mismatch between pre-training dataset and fine-tuning dataset is the main cause for hallucinations is gaining traction from prior works. To reduce such effect, we introduce Continual Pre-Training for Hallucinations (CPTHalu), a method that performs fine-tuning of a sample in parallel with continued pre-training of its corresponding factual knowledge. We formulate a novel RL framework for promoting factual consistency of the generated answers. Specifically, GRPO was leveraged to reward word-level F1-scores and penalize output length differences. We adapt GRPO to reading comprehension via our training scheme, a first effort for RL to perform knowledge fine-tuning in reading comprehension to our understanding. Our experiments on HaluEval and SQuAD obtain large and consistent performance increases of up to 17 points. To further assess factual grounding, we also perform ablation study with our new Augmented QA benchmarks, being novel question-answer pairs over the same source documents. We obtain improvements for both closed-book and open-book performance. We also validate scalability on smaller models, showing that CPTHalu's benefits persist under limited capacity. Our results establish CPTHalu as a simple yet effective strategy for mitigating hallucinations in LLMs. Our code and dataset will be released upon publication.

## 1 INTRODUCTION

Large language models (LLMs) are increasingly deployed in user-facing assistants and developer tools, yet their tendency to hallucinate — to produce content that contradicts the source evidence or cannot be verified by it — limits reliability in practice. Addressing hallucination is therefore central to safer, more dependable LLM applications.

A growing body of work attributes a substantial portion of hallucinations to distribution mismatch (Kang et al., 2025; Gekhman et al., 2024) between the data used for pre-training and the data encountered during fine-tuning and evaluation. When examples fall outside the model's familiar pre-training distribution, errors proliferate even if the model is ostensibly competent at the task. Motivated by this hypothesis, we propose CPTHalu, a simple concurrent training recipe that reduces this mismatch by continually pre-training on the very knowledge paragraphs used by reading-comprehension tasks while the model is being optimized for answer quality via reinforcement learning (RL).

For model fine-tuning, we adopt GRPO (Shao et al., 2024), with the reward function defined based on the word-level F1 score. Empirical results show that the training process is stable and does not exhibit any notable inconsistencies. To the best of our knowledge, this work is the first to introduce GRPO as a methodology for knowledge-centric fine-tuning. By leveraging GRPO in this context, we aim to enhance the model's ability to retain and apply factual information more reliably.

We study CPTHalu on HaluEval and SQuAD, two reading-comprehension benchmarks that supply a question, an answer, and a supporting paragraph per instance. Across both datasets, our GRPO baseline already delivers large absolute gains over a supervised baseline (e.g., +17 EM on SQuAD and +9 EM on HaluEval), and CPTHalu provides additional, consistent improvements on top of GRPO with

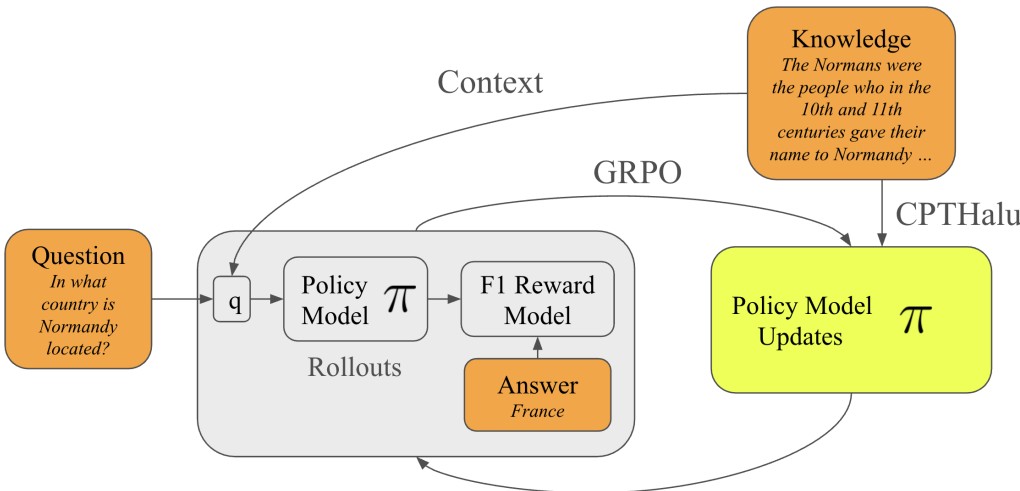

Figure 1: Our methodology. Grey area is the RL process and orange areas are components of reading comprehension task, being question, answer and knowledge. Yellow area depict loss construction and subsequent model updates. The continual pre-training loss $L_{\text{CPT}}$ predicts tokens of the knowledge paragraph $k$ in parallel with GRPO policy loss; the joint objective is $L = L_{\text{GRPO}} + \alpha L_{\text{CPT}}$ (Eq. 6).

stable training dynamics. To test whether the concurrent rehearsal strengthens document-level factual grounding rather than merely rewarding lexical overlap, we further introduce Augmented Q&A evaluations that hold the paragraph fixed but replace each original question–answer pair with new ones; CPTHalu improves performance in both No Context (closed-book) and Knowledge Context (open-book) settings.

Additionally, we examine a small-model setting (Qwen-2.5 0.5B) to probe scalability and reward design. We find that straightforward reward shaping (word-level F1 plus a length-constrained reward) yields gains and that CPTHalu's benefit persists at reduced scale, suggesting the approach is not confined to a particular backbone or compute regime.

Our contributions are the following:

1. Our findings indicate that continuous pre-training consistently mitigates model hallucinations. We evaluate our models on HaluEval (Li et al., 2023) and SQuAD (Rajpurkar et al., 2016) dataset and report performance increases in reading comprehension settings.

2. We demonstrate that GRPO constitutes a robust framework for optimizing models with respect to factuality scores. We utilize GRPO to optimize our LLMs with word-wise F1 scores as rewards.

3. We conduct ablation studies that incorporate evaluations using an augmented QA dataset. We design an augmented QA dataset that utilizes the same knowledge documents as HaluEval, but with different question-answer pairs. The purpose of the augmented QA dataset is to verify whether the models remember knowledge documents well.

## 2 METHODOLOGY

### 2.1 GRPO

In our approach, GRPO is applied to the reading comprehension task, but it is not intended to induce or promote reasoning. While prior studies (Shao et al., 2024; DeepSeek-AI, 2025) have employed GRPO in domains such as complex mathematical problem solving, which inherently require strong reasoning abilities, our work adopts a simpler reward design focused on improving factuality.

Specifically, GRPO leverages rewards based on word-level F1 scores and output length differences, guiding the model to maintain consistency between the knowledge paragraph and the answer.

Figure 1 presents our methodology. GRPO is performed concurrently with Continous Pre-Training for Hallucination reduction (CPTHalu). For GRPO, we utilize 2 reward terms, one being word-wise F1 and the another being length differences.

For word-wise F1, we utilize same logic as original SQuAD (Rajpurkar et al., 2016) and compute based on word overlaps between predicted words and gold words.

$$\text{precision} = \frac{\text{overlap}}{\text{pred\_words}}, \text{recall} = \frac{\text{overlap}}{\text{gold\_words}} \tag{1}$$

$$\text{F1} = \frac{2 * \text{precision} * \text{recall}}{\text{precision} + \text{recall}} \tag{2}$$

"Overlap", "pred_words", and "gold_words" reflect the count of overlapping words in prediction and gold answer, the count of all words in the prediction and the count of all words in the gold answer respectively.

For length difference reward, absolute difference between lengths of prediction and gold answer is calculated. If the difference is 0, reward of 1.0 is given. Else if the difference is same or smaller than 5, reward of 0.5 is given. Else if the difference is same of smaller than 10, reward of 0.2 is given. If the difference is bigger than 10, no reward is given. The length difference reward is provided to protect the model from falling into certain failure modes, such as too long or too short output predictions.

This reward will go through computing RL policy loss via advantage calculation, as per GRPO recipe. Let's depict the loss term for RL as $L_{\text{RL}}$. Then:

$$L_{\text{RL}} = L_{\text{GRPO}}(R_{F1} + R_{\text{length}}) \tag{3}$$

with $R_{F1}$ being reward based on $F1$ score and $R_{\text{length}}$ being reward based on length.

## 2.2 CPTHALU

CPTHalu utilizes an additional loss term to compute the pretraining loss. This term "reads" through the context knowledge and produces a loss value, based on next token prediction scheme.

**Continual pre-training loss** $L_{\text{CPT}}$ Let the knowledge paragraph be tokenized as $k = (k_1, \ldots, k_T)$, and let $\pi_\theta$ denote the autoregressive policy model. We define the continual pre-training term as the standard teacher-forced next-token negative log-likelihood (average cross-entropy) computed *only* on the knowledge tokens:

$$L_{\text{CPT}}(\theta) = \mathbb{E}_{(k,q,a)\sim\mathcal{D}} \left[ -\frac{1}{T} \sum_{t=1}^{T} \log \pi_\theta\big(k_t \mid k_{<t}\big) \right]. \tag{4}$$

In practice, for a mini-batch $\mathcal{B} = \{(k^{(i)}, q^{(i)}, a^{(i)})\}_{i=1}^{B}$, we use the unbiased estimator

$$\widehat{L}_{\text{CPT}} = -\frac{1}{B} \sum_{i=1}^{B} \frac{1}{|k^{(i)}|} \sum_{t=1}^{|k^{(i)}|} \log \pi_\theta\big(k_t^{(i)} \mid k_{<t}^{(i)}\big), \tag{5}$$

masking out all tokens not belonging to the knowledge paragraph. Gradients from $L_{\text{CPT}}$ are combined with the GRPO policy gradients in the joint objective:

$$\widehat{L} = \widehat{L}_{\text{GRPO}}(R_{F1} + R_{\text{length}}) + \alpha \widehat{L}_{\text{CPT}} \tag{6}$$

With $0.0 \leq \alpha \leq 1.0$ denoting the relative importance of $\widehat{L}_{\text{CPT}}$ term.

| role | content |
|------|---------|
| system | This is a conversation between a user and an assistant.

- The assistant provides the answer directly.
- No discussion or thinking process is needed.
- Only the answer should be provided.

**Example**
```
<\|im_start\|>user
Natalia sold clips to 48 of her friends in April,
and then she sold half as many clips in May.
How many clips did Natalia sell altogether in April and May?
<\|im_end\|>
<\|im_start\|>assistant
72
<\|im_end\|>
```

You are an AI assistant that answers a question based on the provided knowledge.

**Knowledge**
```
The Normans (Norman: Nourmands; French: Normands; Latin: Normanni)
were the people who in the 10th and 11th centuries gave their name to
Normandy, a region in France. They were descended from Norse
....
``` |
| user | In what country is Normandy located? |

Table 1: Sample prompt for a prediction. We utilize 1-shot example to force the model to output answer to the question only. "Knowledge" section is removed for "No Context" experiment in Table 5.

## 2.3 AUGMENTED Q&A DATASET

To directly test whether CPTHalu helps models internalize and reuse factual content from the same source documents seen during training, we build augmented Q&A sets by re-using the knowledge paragraphs provided by HaluEval (and analogously SQuAD) while creating entirely new question–answer pairs that do not appear in the original benchmarks. Please see Fig. 2.

Concretely, for each knowledge paragraph $k$, we generate one or more novel questions $q$ whose answers $a$ are short, extractive spans (or minimal paraphrases) supported by $k$. The augmented sets are used only for evaluation; models are trained on the original benchmark splits, while CPTHalu's continual pre-training term is applied to the same knowledge text $k$ to rehearse facts.

The augmented items form an evaluation-only set. They share the same source paragraphs as the training corpus by design (to test knowledge retention), but their question–answer pairs are held out.

## 3 EXPERIMENTAL SETUP

We experiment with the Qwen 2.5 3B Instruct model (Qwen et al., 2025) and utilize the RLYX framework (Kim, 2025) for RL (GRPO) purposes. We utilize 2 nodes, 8 H100 GPUs per node. We train the models in bfloat16 format and utilize Ray Serve for node-to-node communications. We train for 3 epochs, both for the HaluEval dataset (10K samples, 8K train set) and for the truncated SQuAD dataset (10K train set and 10.6K validation set). We save the model checkpoints every 160 steps and report the performance on the checkpoints with the best performance on the validation set.

| role | content |
|------|---------|
| user | You are a helpful assistant.
Ingest the given knowledge and produce a question-answer pair
according to the style of given example,
while concerning different concept from the example.

**Knowledge**
```
The Normans (Norman: Nourmands; French: Normands; Latin: Normanni)
were the people who in the 10th and 11th centuries gave their name to
Normandy, a region in France. They were descended from Norse
....
```

**Example**
[
{ "question": "In what country is Normandy located?"
"answer": "France"}, ....
]
Generate a question-answer pair that differ from the example,
while being stylistically similar. |

Table 2: Sample prompt for generating a pair of augmented Q&A dataset.

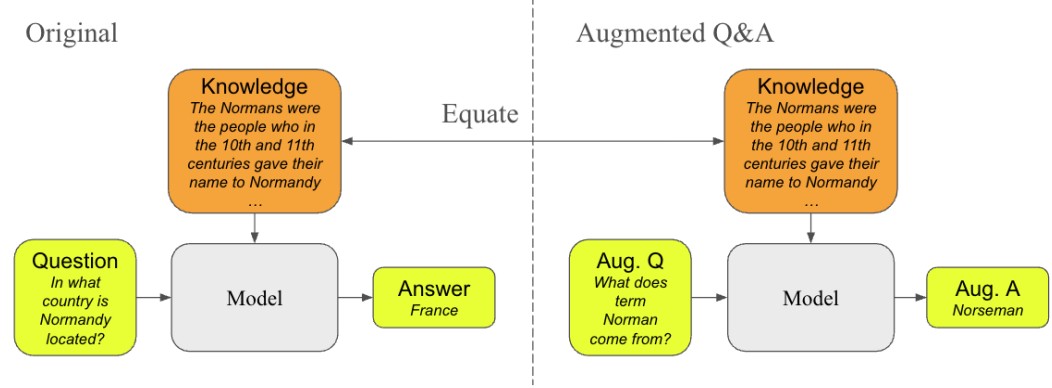

Figure 2: Augmented Q&A data. Grey area is hypothetical model and orange area is knowledge, being kept the same in augmented Q&A data generation process. Augmented Question & augmented Answer (yellow in the right diagram) are generated via an LLM (GPT-4o (OpenAI, 2024)) with respect to the identical knowledge. Please see Table 2 for prompt example.

We set train batch size to be 8 and evaluation batch size to be 64. We set rollout per sample to 3, rollout temperature to be 1.0, and rollout maximum tokens to be 500. Learning rate is $5e^{-7}$ with cosine scheduler type and maximum gradient value of 1.0. Our prompt is described in the following Table 1.

## 4 RESULTS

### 4.1 OVERALL RESULTS

We display our average Exact Match (EM) and F1 scores in Table 3. We report up to 17 point and 10 point EM increase for SQuAD and HaluEval respectably. We also report 14 point and 7

| Model | HaluEval | | SQuAD | |
|---|---|---|---|---|
| | EM | F1 | EM | F1 |
| Baseline | 73.10 ($\pm$0.00) | 82.09 ($\pm$0.00) | 53.08 ($\pm$0.02) | 70.73 ($\pm$0.01) |
| GRPO | 82.17 ($\pm$0.29) | 89.14 ($\pm$0.15) | 70.01 ($\pm$0.12) | 84.03 ($\pm$0.18) |
| CPTHalu 0.1 + GRPO | 82.65 ($\pm$0.17) | 89.27 ($\pm$0.17) | **70.48** ($\pm$0.27) | 84.31 ($\pm$0.26) |
| CPTHalu 1.0 + GRPO | **83.10** ($\pm$0.10) | **89.51** ($\pm$0.12) | 70.14 ($\pm$0.12) | **84.34** ($\pm$0.46) |

Table 3: Exact match and F1 results on HaluEval and SQuAD dataset.

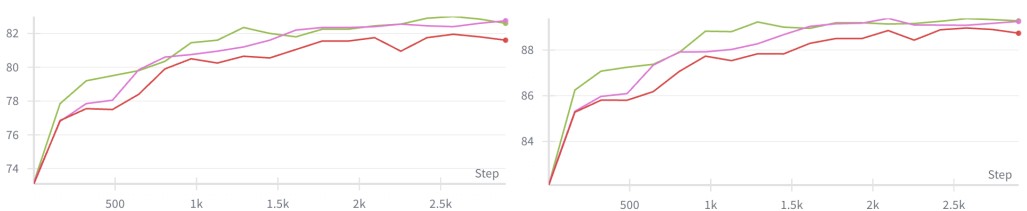

Figure 3: HaluEval train results on the validation set. Left is Exact Match (EM) and right is F1 score. Red is without CPTHalu, pink is CPTHalu with $\alpha = 0.1$, and green is CPTHalu with $\alpha = 1.0$.

point F1 score increase for SQuAD and HaluEval. Our results show that CPTHalu is effective, with consistent 1 point EM increase in both datasets, compared to GRPO only.

## 4.2 HALUEVAL RESULTS

We display training results on HaluEval, evaluated on validation set (Table 3, Figure 3). We observe consistent performance increases, both for CPTHalu with $\alpha = 0.1$ and $\alpha = 1.0$, in contrast to without CPTHalu. $\alpha = 1.0$ emerges as the best HaluEval setting (83.10 EM / 89.51 F1). We also observe that training process is stable without catastrophic forgetting.

## 4.3 SQUAD RESULTS

We display training results on SQuAD, evaluated on validation set (Table 3, Figure 4). We observe that adding CPTHalu increases the performance of the models during the training process. CPTHalu's curves remain closely above GRPO, indicating faster progress toward the final plateau.

## 4.4 SMALL MODEL RESULTS

We experiment with Qwen 2.5 0.5B model with various reward configurations and report the results in Table 4, Figure 5. We report similar EM values, while F1 score increases more than 1 point. Reward shaping matters, with F1 scores increasing from 81.42 to 82.73 as we add new terms to the mix (F1, length and CPTHalu). We also observe that CPTHalu's benefit is visible even at small scale.

## 4.5 AUGMENTED HALUEVAL Q&A RESULTS

Definition to Augmented HaluEval Q&A dataset is in Section 2.3. To probe whether the model better internalizes facts from the same documents rehearsed during training, we evaluate on Augmented HaluEval under No Context (closed-book) and Knowledge Context (open-book) settings. As shown in Table 5, we find that performance increases for both views. "No context" is the experiment without "Knowledge" found in system prompt of Table 1. This indicates that the CPTHalu-trained models understand training set knowledge, and uses it to generate answers to given questions.

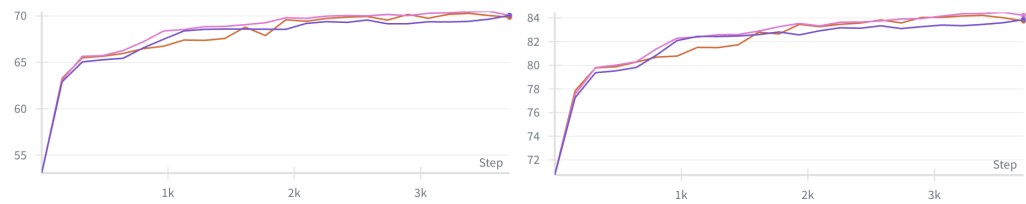

Figure 4: SQuAD train results on the validation set. Left is Exact Match (EM) and right is F1 score. Purple is without CPTHalu, pink is CPTHalu with $\alpha = 0.1$, and red is CPTHalu with $\alpha = 1.0$.

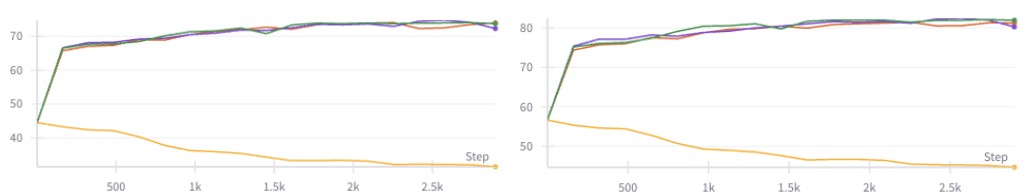

Figure 5: 0.5B train results for HaluEval on the validation set. Left is Exact Match (EM) and right is F1 score. Yellow is pretraining only without GRPO, green is F1 reward only without length reward, purple is with both F1 reward and length reward, and red is EM reward only without F1 or length reward. See Section 4.4 for analysis. Performance captured in Table 4.

### 4.6 Augmented SQuAD Q&A Results

Definition to Augmented SQuAD Q&A dataset is in Section 2.3. As shown in Table 6, we find that performance increases by more than 1 point for EM with CPTHalu 0.1, compared to no CPTHalu case. In Knowledge Context, $\alpha = 0.1$ gives the best result, following the trend in Table 3.

For augmented SQuAD Q&A the performances are overall lower, which may be due to the fact multiple questions are already provided per train set documents, compared to HaluEval where only 1 question is provided. This may cause the LLM to come up with harder questions than HaluEval. Overall, the results show that CPTHalu is able to teach the models training set knowledge. The models then uses such knowledge to generate answers to questions.

## 5 Analysis

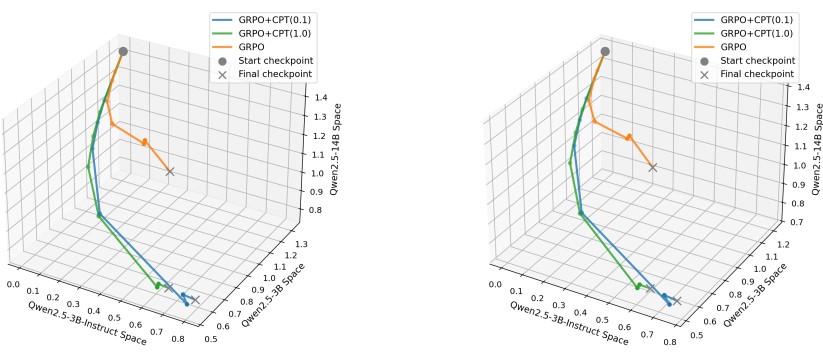

Figure 6: Average symmetric KL divergence distances (Eq. 10). Left is train documents and right is test documents. The models start from grey circle points, undergoes training with our loss (Eq. 6), and ends up in grey X points. Interpretation in Section 5.

| Reward | EM | F1 |
|---|---|---|
| Pretrain Only | 44.50 | 56.63 |
| EM Only | 74.00 | 81.42 |
| F1 Reward | 74.00 | 82.10 |
| F1 + Length Reward | 74.70 | 82.42 |
| CPTHalu 0.1 + GRPO | 74.25 | 82.63 |
| CPTHalu 1.0 + GRPO | **74.75** | **82.73** |

Table 4: Exact match and F1 results on 0.5B model. Description in Section 4.4. Training process captured in Figure 5.

| Model | No Context | | Knowledge Context | |
|---|---|---|---|---|
| | EM | F1 | EM | F1 |
| GRPO | 16.89 ($\pm$0.57) | 24.13 ($\pm$0.69) | 76.86 ($\pm$1.14) | 86.81 ($\pm$0.84) |
| CPTHalu 0.1 + GRPO | 17.38 ($\pm$0.64) | 24.71 ($\pm$0.58) | **77.46** ($\pm$0.26) | **87.40** ($\pm$0.18) |
| CPTHalu 1.0 + GRPO | **17.55** ($\pm$0.81) | **25.02** ($\pm$0.57) | 77.04 ($\pm$0.80) | 87.11 ($\pm$0.67) |

Table 5: Exact match and F1 results on Augmented HaluEval Q&A. Please find the description for the dataset at Section 4.5.

We perform probabilistic analysis with 1000 knowledge paragraphs from HaluEval. We obtain token probabilities per paragraphs and compute distance on how the token-wise probabilities change with relative to original baselines, from pretrained model and instruct-tuned model. We find that CPTHalu models end up being more far similar to pretrained model, in comparison with non-CPTHalu baseline model, achieving up to 50% distance decreases with final checkpoints. This may be due to jointly optimizing for CPTHalu and GRPO, probability distribution getting closer to pretrained models due to continual pre-training loss (Eq. 4) while drifting farther away from instruct models via reinforcement learning (Eq. 3). Distance between 14B model also decreases, signaling that our model reproduces "smarter" model's token-wise probabilities.

### 5.1 SYMMETRIC KL DIVERGENCE SPACE

We compute symmetric KL divergence via following mechanism. First, we start with the definition for KL divergence, as expressed below:

$$\text{KL}(p \parallel q) = \sum_i p_i \log \frac{p_i}{q_i}. \tag{7}$$

$p_i$ being tokenwise probability for the given model, and $q_i$ being tokenwise probability for a reference model (Qwen 2.5 3B and Qwen 2.5 3B Instruct in our case).

To have a distance that's symmetric, we should change asymmetric KL divergence and compute 2 values with different baselines.

$$D_t^{p\|q} = \text{KL}\big(p^{(t)} \parallel q^{(t)}\big), \qquad D_t^{q\|p} = \text{KL}\big(q^{(t)} \parallel p^{(t)}\big). \tag{8}$$

We should then average such KL divergence per tokens to be generated.

$$\overline{\text{KL}}(p \parallel q) = \frac{1}{T} \sum_{t=1}^{T} D_t^{p\|q}, \qquad \overline{\text{KL}}(q \parallel p) = \frac{1}{T} \sum_{t=1}^{T} D_t^{q\|p}. \tag{9}$$

With $T$ being length of tokens to be generated.

Lastly, we take average of two distinct KL divergence values and define them as "Symmetric KL Divergence distance".

| Model | No Context | | Knowledge Context | |
| --- | --- | --- | --- | --- |
| | EM | F1 | EM | F1 |
| GRPO | 5.44 | 14.81 | 58.09 | 80.15 |
| CPTHalu 0.1 + GRPO | **6.59** | 15.47 | **59.18** | **80.95** |
| CPTHalu 1.0 + GRPO | 6.46 | **15.70** | 58.86 | 79.99 |

Table 6: Exact match and F1 results on Augmented SQuAD Q&A. Please find the description for the dataset at Section 4.6.

$$D_{\mathrm{sym}}(p, q) = \tfrac{1}{2}\Big(\overline{\mathrm{KL}}(p \parallel q) + \overline{\mathrm{KL}}(q \parallel p)\Big). \tag{10}$$

$D_{\mathrm{sym}}$ is the distance value we use to graph Fig. 6.

## 6 RELATED WORKS

Prior studies suggest that hallucinations often arise from mismatches between the knowledge distributions of fine-tuning and pre-training datasets. (Kang et al., 2025; Gekhman et al., 2024). We take inspiration from their findings and add a concurrent pre-training stage to fine-tuning setup, reducing hallucinations and improving the model performance.

GRPO (Shao et al., 2024) is known to work with solving hard mathematics problems. While there exists a few works that handle knowledge with RL (Li et al., 2025; Xu et al., 2025) they are either on an entirely different domain (visual relation comprehension) or on a mathematics dataset. Our work is the first to utilize GRPO on reading comprehension dataset (Rajpurkar et al., 2016), and we achieve a performance increase of 17% on SQuAD.

HaluEval (Li et al., 2023) and SQuAD (Rajpurkar et al., 2016) are reading comprehension datasets that provide a knowledge paragraph per question-answer pairs. We differ from such datasets in that we create a new question-answer pair that is not present in such datasets, instead we augment such datasets based on the provided knowledge. We test on this additional data such that we further evaluate reading comprehension on the knowledge we visited during continual pre-training.

Fu et al. (2025) also explores single-stage integration of SFT and RL for reasoning tasks. For example, SRFT unifies supervised fine-tuning and reinforcement learning through entropy-aware weighting, showing that SFT induces coarse-grained global policy shifts while RL makes fine-grained selective adjustments. Their framework, tested on mathematical reasoning benchmarks, demonstrates that integrating both paradigms in one stage can improve efficiency and stability. In contrast, our work does not aim to induce chain-of-thought reasoning. Instead, we adopt a factuality-oriented reward design (word-level F1 and output length prior) and couple it with continual pre-training on evidence paragraphs, directly addressing hallucinations in reading comprehension. This positions our method as complementary to SRFT: whereas SRFT balances global and selective updates for reasoning, our approach uses continual pre-training to supply global factual grounding while GRPO selectively optimizes answer quality in knowledge-centric settings.

## 7 CONCLUSION

We introduce CPTHalu, a simple concurrent training recipe that couples reinforcement learning with continual pre-training on the same evidence paragraphs used by reading-comprehension tasks. Formally, CPTHalu optimizes a joint objective that adds a next-token prediction loss on the knowledge paragraph to the GRPO policy loss. This design directly targets the pretrain–finetune mismatch hypothesized to underlie many hallucinations by rehearsing document-level facts while optimizing answer quality. We believe CPTHalu is a robust hallucination prevention paradigm for reinforcement learning training.

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
