# OpenReview forum: "Continual Pre-Training for Hallucination Reduction"
_ICLR.cc/2026/Conference — Submitted to ICLR 2026_

### Official Review · Reviewer_j5iK · 2025-10-16

**Soundness:** 2
**Presentation:** 2
**Contribution:** 2
**Rating:** 2
**Confidence:** 4

**Summary:**

The paper proposes CPTHalu, a joint training recipe that runs GRPO-based RL for answer quality alongside continual pre‑training (next‑token prediction) on the same evidence paragraph.

Evaluated on HaluEval and SQuAD, GRPO already yields large gains over a supervised baseline, and adding CPTHalu provides consistent but modest additional improvements. Augmented Q&A evaluations (closed/open book) and a 0.5B model study show similar trends.

**Strengths:**

Simple, general recipe: Clear joint loss (GRPO + continual pre‑training) that is architecture‑agnostic and easy to integrate.

Stable training: Plots show smooth learning dynamics; no catastrophic forgetting reported.

Breadth of evaluation: Standard RC datasets (HaluEval, SQuAD), augmented Q&A to probe knowledge retention, and small‑model (0.5B) scaling analysis with reward ablations.

**Weaknesses:**

Incremental gains over GRPO: While the paper highlights “up to 17 EM” on SQuAD, that jump is largely GRPO vs. supervised; CPTHalu typically adds ~0–1 EM over GRPO (e.g., HaluEval 82.17→83.10 EM; SQuAD 70.01→70.48/70.14). The marginal benefit may be statistically small.

Hallucination proxy: Uses EM/F1 on reading comprehension as the main signal; lacks direct hallucination/faithfulness metrics or human judgments, so the link to “hallucination reduction” is indirect.

Rehearsal risk: Continual pre‑training on the very paragraphs used for QA may blur the line between grounding and memorization; augmented Q&A gains in closed‑book are small (e.g., HaluEval EM 16.89→17.55).

Augmented data quality: The augmented Q&A is generated by an external LLM (GPT‑4o); the paper provides limited validation/quality‑control details (e.g., deduping, paraphrase checking, manual auditing).

Missing baselines/ablations at 3B: A clear pretrain‑only vs GRPO‑only vs CPTHalu‑only comparison is shown for 0.5B, but not for the main 3B model; comparisons to alternative RL/DPO/RLAIF or retrieval‑augmented baselines are absent.

Reward design is heuristic: Length‑difference reward uses coarse thresholds; sensitivity analyses (scales, normalization) and α\alphaα selection are limited.

**Questions:**

Measurement of hallucination: Beyond EM/F1, did you compute any faithfulness or hallucination‑focused metrics (or run human evaluations) to substantiate the claim of hallucination reduction?

Ablations at 3B: What are the results for pretraining‑only and CPTHalu‑only (no GRPO) at 3B to isolate each component’s contribution?

Augmented Q&A validation: How were items filtered for novelty vs. paraphrase, deduplicated, and quality‑checked? Will you release the augmented sets with provenance/validation metadata?

Sensitivity & scaling: How sensitive are outcomes to α\alphaα, reward weights, and rollout parameters? Any normalization/adaptive weighting between LGRPOL_{\text{GRPO}}LGRPO​ and LCPTL_{\text{CPT}}LCPT​?

Generalization: Does continual pre‑training on training paragraphs affect performance on unseen domains or retrieval‑based QA? Any results on out‑of‑paragraph grounding tasks?

Cost/variance: What is the compute/time overhead of CPTHalu vs. GRPO‑only, and how robust are results across multiple seeds/checkpoints (statistical significance)?

---

### Official Review · Reviewer_B3cV · 2025-10-26

**Soundness:** 3
**Presentation:** 3
**Contribution:** 2
**Rating:** 4
**Confidence:** 3

**Summary:**

The paper proposes to simultaneously fine-tune and (continually) pre-train Large Language Models (LLMs) to reduce hallucinations caused by mismatch between pre-training and fine-tuning data.

**Strengths:**

The paper validates to a certain extent the hypothesis that hallucinations of LLMs are caused by mismatch between pre-training and fine-tuning data, even though its implications may be limited (more on this in “Weakness”).

**Weaknesses:**

Although the paper sheds more light on the possible cause of hallucinations, it may or may not imply a general hallucination mitigation strategy. Intuitively, mismatches between pre-training and fine-tuning data should be crucial for LLMs to generalize. Thus the impact of the proposed technique on the general capabilities of LLMs should also be studied.

**Questions:**

Please see “Weakness”.

---

### Official Review · Reviewer_7qMP · 2025-10-29

**Soundness:** 2
**Presentation:** 2
**Contribution:** 2
**Rating:** 4
**Confidence:** 3

**Summary:**

This paper introduces CPTHalu, a concurrent training framework designed to mitigate hallucinations in large language models (LLMs).
The approach combines GRPO-based reinforcement learning (RL) with continual pre-training on the same factual paragraphs used during fine-tuning. By jointly optimizing the GRPO policy loss and a continual pre-training loss term
L = L_GRPO + α L_CPT, the method aims to reduce the pre-trained/fine-tune distribution mismatch that contributes to hallucination. Experiments on HaluEval and SQuAD show consistent improvements (up to +17 EM and +14 F1), and further validation on small models and augmented Q&A datasets demonstrates factual retention and robustness.

**Strengths:**

1. Methodological clarity. The formulation is explicit, including reward design (word-level F1 and output-length constraints) and its adaptation of GRPO to factuality rather than reasoning.
2. Analytical insight. The symmetric KL-divergence analysis provides a probabilistic interpretation of why CPTHalu models stay closer to the pretraining distribution.

**Weaknesses:**

1. Limited significance of gains. Improvements of roughly +1 EM/F1 over GRPO are modest; further statistical validation is needed.
2. Missing baselines. Comparison with other factuality-enhancing approaches (e.g., KDRL, retrieval-augmented fine-tuning) would clarify relative strengths.
3. Possible data bias. Augmented Q&A sets are generated using GPT-4o, potentially introducing stylistic or lexical bias.

**Questions:**

1. Does continual pre-training risk catastrophic forgetting of instruction-following ability?
2. Can CPTHalu generalize to other tasks such as summarization or open-ended generation?
3. Could this framework integrate with retrieval or knowledge-editing systems to further mitigate hallucination?
4. In the KL-divergence analysis, which models define p and q? Clarify what “closer to the pre-trained model’’ means quantitatively.

---

### Official Review · Reviewer_mp5E · 2025-11-01

**Soundness:** 3
**Presentation:** 2
**Contribution:** 2
**Rating:** 4
**Confidence:** 2

**Summary:**

The paper introduces CPTHalu, a method designed to reduce hallucinations by combining reinforcement learning fine-tuning with continual pre-training on the same knowledge used for reading comprehension tasks. The authors hypothesize that hallucinations often stem from a "distribution mismatch" between a model's pre-training data and its fine-tuning data. Using the GRPO algorithm with rewards based on word-level F1 and output length, CPTHalu simultaneously reinforces factual accuracy while maintaining training stability. Experiments on HaluEval and SQuAD show consistent performance gains. The approach also generalizes across model sizes and improves factual grounding in augmented question–answer datasets, which uses the same documents but new questions.

**Strengths:**

- Presents a integration of continual pre-training with RL (combining GRPO with a masked next-token loss on the evidence paragraph), directly addressing the pretrain–finetune distribution mismatch that leads to hallucinations.
- Demonstrates consistent empirical results on multiple benchmarks with measurable factuality improvements.
- Includes detailed ablation and augmented Q&A studies, providing thorough validation of the method’s factual grounding capability. augmented Q&A to test whether models internalize document facts rather than just match patterns, with improvements reported in both No-Context and Knowledge-Context views.

**Weaknesses:**

- Incremental benefit of CPTHalu over GRPO is modest, typically around 0.5–1 EM and fractions of a point in F1, which may limit practical impact relative to the RL baseline alone. While CPTHalu consistently outperforms the GRPO-only baseline, the additional performance gain from the CPTHalu component is relatively small. The gain is mostly attributable to the GRPO baseline itself, not the novel CPTHalu addition
- The reward design, while effective, relies on simplistic metrics (F1 and length) that may not fully capture nuanced factual correctness. Such reward design optimizes surface-level word overlap and output length, which may bias models toward extractive phrasing without directly measuring faithfulness or reasoning.
- The primary experiments are conducted on a single model family,  the findings are not validated on other common architectures (like Llama or Mistral). Some analysis, such as the KL divergence interpretation, remains qualitative and lacks deeper theoretical justification for observed improvements.
The CPTHalu method is inherently tied to tasks where a specific "knowledge paragraph" is available during training. This work does not address the broader, more common problem of parametric hallucinations, where a model invents facts in an open-domain setting without any provided source text.

**Questions:**

- Given the marginal gains over the strong GRPO baseline, is the added computational complexity of CPTHalu practically justified?
- Does optimizing for F1 and length rewards encourage true factual faithfulness, or does it merely bias the model toward better extractive phrasing?
- Since experiments were limited to the Qwen family, how do we know if the CPTHalu method generalizes to other model architectures like Llama or Mistral?
- As CPTHalu requires a "knowledge paragraph" for training, how could this method be adapted to reduce parametric hallucinations in open-domain settings that lack explicit source text?

---

### Meta-Review · Area_Chair_4iNz · 2026-01-03

**Summary:**

All reviews are negative about the work.

In summary, most reviewers express concerns regarding 1) the significance of the method and empirical improvements, 2) the lack of enough justifcation of underlying mechanisms and effectivness of generalization, 3) the limited number of models/datasets used for experiments.

With all those concerns remain unadressed, the submissiion would benefit from another round of major revision.

**Reviewer Concerns:**

The authors did not provide any response.

**Reviewer Scores:**

The scores would remain the same since the authors did not provide any response.

---

### Decision · Program_Chairs · 2026-01-26

Reject